# EEG-X: Device-Agnostic and Noise-Robust Foundation Model for EEG

## Abstract

Foundation models for EEG analysis are still in their infancy, limited by two key challenges: (1) variability across datasets caused by differences in recording devices and configurations, and (2) the low signal-to-noise ratio (SNR) of EEG, where brain signals are often buried under artifacts and non-brain sources. To address these challenges, we present *EEG-X*, a device-agnostic and noise-robust foundation model for EEG representation learning. EEG-X introduces a novel location-based channel embedding that encodes spatial information and improves generalization across domains and tasks by allowing the model to handle varying channel numbers, combinations, and recording lengths. To enhance robustness against noise, EEG-X employs a noise-aware masking–reconstruction strategy in both raw and latent spaces. Unlike previous models that mask and reconstruct raw noisy EEG signals, EEG-X is trained to reconstruct denoised signals obtained through an artifact removal process, ensuring that the learned representations focus on neural activity rather than noise. To further enhance reconstruction-based pretraining, EEG-X introduces a novel dictionary-inspired convolutional transformation (DiCT) layer that projects signals into a structured feature space before computing reconstruction (MSE) loss, reducing noise sensitivity and capturing frequency- and shape-aware similarities. Experiments on datasets collected from diverse devices show that EEG-X outperforms state-of-the-art methods across multiple downstream EEG tasks and excels in cross-domain settings where pre-trained and downstream datasets differ in electrode layouts. The models and code are available at here.

## 1 Introduction

Electroencephalography (EEG) is a non-invasive method for recording brain activity by placing electrodes on the scalp, offering high temporal resolution and relatively low cost (Teplan et al., 2002; Müller-Putz, 2020). It is widely used in both neuroscience and clinical practice, where it supports the diagnosis of neurological disorders such as epilepsy and sleep disorders (Lotte et al., 2018; Boonyakitanont et al., 2020), as well as in applications like sleep stage classification, affective state analysis, motor imagery decoding, stress detection, and seizure classification (Craik et al., 2019; Hosseini et al., 2020). Despite these benefits, effectively modeling EEG data often requires large amounts of annotated data, expert knowledge, and techniques specially designed to the unique characteristics of each task (Weng et al., 2024).

Recently, self-supervised pretraining has achieved remarkable success in natural language processing (NLP), computer vision (CV), and speech/audio processing (Bommasani et al., 2021; Yu et al., 2022; Chen et al., 2024). Models pretrained on large unlabeled datasets can be efficiently adapted to a wide range of downstream tasks, often requiring little or no labeled data. A major advantage of these models is their ability to handle multiple tasks within a single framework, unlike traditional approaches that typically require large labeled datasets and task-specific algorithms. This representation learning approach simplifies development by reducing the need for task-specific models and extensive labeling, while also improving generalization

by leveraging shared patterns from diverse data. This makes the model more adaptable and efficient across various tasks, forming the basis of what is known as a foundation model.

While foundation models have shown great success in vision, language, and speech, their application to EEG data remains limited and has yet to achieve comparable breakthroughs (Weng et al., 2024; Moham-madi Foumani et al., 2024b). Two key challenges hinder the broader adoption of EEG foundation models: (1) substantial variability across EEG datasets due to differences in recording equipment, which introduce variations in sampling frequencies, electrode placements, and the number of channels, and (2) the inherently low signal-to-noise ratio of EEG, where neural activity is often buried under artifacts and non-brain sources (Jiang et al., 2019; Pion-Tonachini et al., 2019; Blum et al., 2019). To address these challenges, we propose EEG-X, a foundation model for EEG representation learning that learns device-agnostic and noise-robust representations through a location-based channel embedding to handle variability across devices, a noise-aware masking–reconstruction strategy for robust pretraining, and a dictionary-inspired transformation layer that enhances reconstruction loss by balancing comparisons across frequency bands.

The first component, location-based channel embedding, is designed to handle device variability. Earlier models typically rely on learnable channel embeddings (Yang et al., 2024; Jiang et al., 2024; Wang et al., 2024; 2025) to indicate signal source, which can work within a single dataset but transfer poorly across devices, as the same channels may not exist across different EEG devices.In addition, such embeddings do not account for electrode positions or preserve the similarity between neighboring regions on the scalp, both of which are important for modeling brain activity. To overcome these issues, EEG-X encodes each electrode's spatial location together with its neighbors, ensuring that nearby electrodes receive similar embeddings.

The second component tackles the challenge of low SNR in EEG. EEG-X employs a noise-aware masking–reconstruction strategy in both raw and latent spaces. Unlike prior models that mask inputs and reconstruct the original noisy signals (Kostas et al., 2021; Chien et al., 2022; Wang et al., 2024; 2025), EEG-X reconstructs artifact-removed signals obtained through an artifact removal process, ensuring that the learned representations capture neural activity rather than noise. In latent space reconstruction, the model predicts the representations of unmasked data from masked samples in an abstract space, where irrelevant raw-level details are suppressed.

Finally, to improve reconstruction loss, EEG-X introduces a **Di**ctionary **C**onvolution **T**ransformation (DiCT) layer. While Mean Squared Error (MSE) remains the default reconstruction loss in self-supervised prediction tasks and performs well in high-SNR domains such as vision, language, and speech (Baevski et al., 2023; 2022), it is less effective for EEG time series (Thompson, 1990). First, in noisy signals, MSE is dominated by high-amplitude components and heavily penalizes large errors. Second, MSE does not consider frequency similarities of two time series. Third, MSE for time series does not directly consider or evaluate the "shape" of the series (Please refer to the Appendix B.1 for details and demonstrations). EEG-X mitigates these issues by projecting both artifact-removed inputs and reconstructed output into a structured space using random convolutional kernels with varying dilations, before computing MSE loss. Random convolutions have proven effective feature extractors for time series classification (Dempster et al., 2020; 2023). This transformation reduces noise sensitivity, balances comparisons across frequency bands, and incorporates shape similarity. This transformation can serve as a lightweight plug-in to improve existing reconstruction-based methods. Here is a summary of contributions:

- **Location-Based Channel Embedding**: Encodes electrode positions and their neighborhoods, preserving brain-region similarity and enabling robust transfer across devices with different channel layout.

- **Noise-Aware Reconstruction**: Learns from artifact-removed signals in both raw and latent spaces, ensuring representations focus on neural activity and remain robust to low-SNR EEG.

- **Dictionary Convolution Transformation (DiCT)**: A lightweight random convolutional projection layer that mitigates MSE limitations by reducing noise sensitivity, balancing comparisons across frequency bands, and incorporating shape similarity into reconstruction loss.

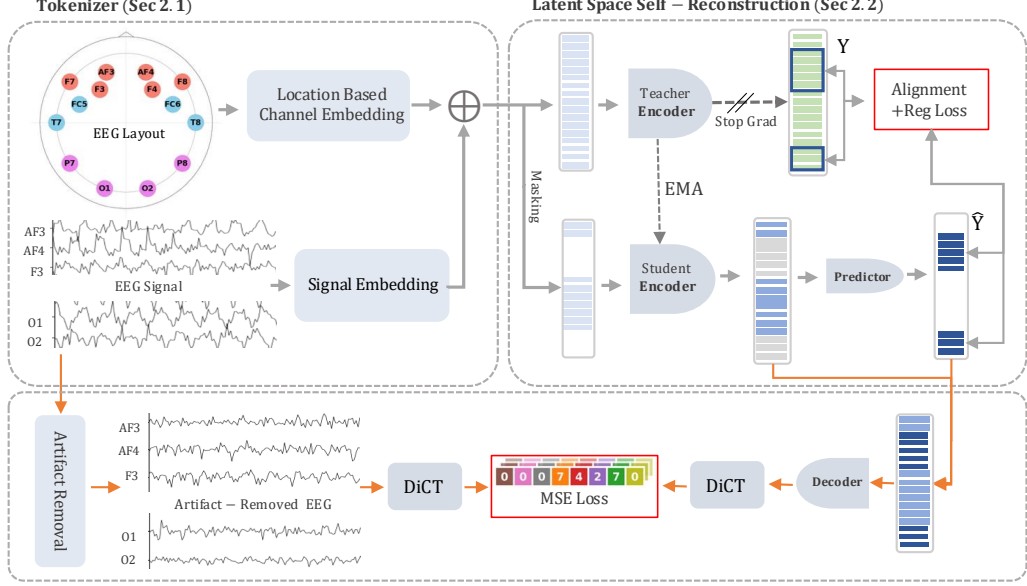

Figure 1: Architecture of EEG-X with three components—tokenizer (with location-based channel embedding), latent space self-reconstruction, and raw space reconstruction with artifact removal and DiCT. Pretrained using latent- and raw-space reconstruction losses.

- **Extensive evaluation**: EEG-X establishes itself as a *foundation model* for EEG, achieving state-of-the-art performance across diverse tasks and datasets, and demonstrating strong cross-domain generalization even when pretraining and downstream datasets differ in tasks and electrode configurations.

## 2 METHODOLOGY OF EEG-X

Fig.1 illustrates the architecture of EEG-X, which consists of three main components: (1) an input tokenizer with location-based channel embedding and signal embedding, (2) latent space self-reconstruction, and (3) noise-aware raw space reconstruction with artifact removal and a DiCT layer. First, the raw EEG signal is divided into equal-sized segments and fed into the signal embedding, producing input tokens. To enhance these tokens with spatial information, we incorporate our novel location-based channel embeddings, derived from electrode locations (Section 2.1). The enriched patches are then processed using a dual self-supervised learning approach: a student–teacher architecture in latent space and an encoder–decoder framework in raw space. Specifically, in latent space, we encode a masked version of the input EEG (student mode) and generate training targets by encoding the unmasked version using an EMA teacher (Section 2.2). In parallel, the raw EEG signal undergoes artifact removal, and both the artifact-removed input and the reconstructed signal are passed through a DiCT layer before computing the reconstruction loss. This complementary objective ensures robust, artifact-free, and semantically rich EEG feature learning (Section 2.3). For downstream tasks, the tokenizer and student encoder are frozen, and only an off-the-shelf classifier is trained on top of the student encoder's representations.

### 2.1 EEG TOKENIZER

Our goal is to model heterogeneous EEG signals—recorded using various devices with different channel configurations, combinations, and recording lengths—within a unified framework.

**Signal Embedding:** Let $X = \{x_1, x_2, \ldots, x_C\}$ represent EEG recordings from $C$ channels, each of length $L$, where $x_c \in \mathbb{R}^L$. To handle varying recording lengths, we tokenize each channel's recording into fixed-length segments of size $w \in \mathbb{R}^+$ with an overlap of $o$ ($o < w$, typically $o = w/4$). Each token $t_{c,i}$ is

generated by sliding a window of size $w$ with overlap $o$ across the signal:

$$t_{c,i} = x_c[(i-1) \cdot (w-o) + 1 : i \cdot w], \quad \text{for} \quad i = 1, 2, \ldots, \left\lceil \frac{L}{w-o} \right\rceil \tag{1}$$

Each token $t_{c,i}$ represents a fixed-length segment of the original signal $x_c$. To capture both temporal and spectral characteristics, we apply the Short-Time Fourier Transform (STFT) to each token, using the magnitude of the frequency representation (Yang et al., 2024). The transformed tokens are then passed through a linear layer to obtain embeddings $e_{c,i}$:

$$e_{c,i} = W \left| \text{STFT}(t_{c,i}) \right| + b, \quad e_{c,i} \in \mathbb{R}^{d_e} \tag{2}$$

Here, $\mathbb{R}^{d_e}$ denotes the embedding dimension, and the transformation is parameterized by a learned weight matrix $W$ and bias $b$. This tokenization process is similar to natural language processing, where each EEG sample is treated as a sequence of tokens, similar to words in a sentence.

While this signal embedding addresses variations in recording length, handling channel variability remains a challenge. Previous approaches train an embedding parameter for all channels in the data and add the corresponding channel embedding to the token (Yang et al., 2024; Jiang et al., 2024; Wang et al., 2024; 2025); however, they present several limitations: (1) The same channels may not exist across different EEG devices, making the learned embeddings less transferable across diverse recording setups. (2) Channel embeddings may not account for the spatial locations of electrodes on the scalp, which are crucial for accurately modeling the brain's spatial activity patterns. (3) They do not preserve the similarity between neighboring regions on the scalp, which can lead to the loss of meaningful spatial relationships. These limitations highlight the need for a new embedding strategy that incorporates both the spatial location of the channels and their relationships across different devices, which our method aims to address.

**Location-based Channel Embedding:** Our goal is to incorporate electrode location information into the model to enhance its representations. The position of an electrode determines the area of the brain whose activity it monitors and hence is critical to interpreting its signal. To encode channel location, we map each electrode to Cartesian coordinates on the scalp using a universal location mapping (Fig.2a). For simplicity, we illustrate the 64-channel system in the figure as the universal location mapping, which outputs coordinates for any input channel. In practice, our mapping is more comprehensive, covering nearly all standard electrode locations on the scalp (see Appendix B.2). To transform these coordinates into embeddings, we adapt sinusoidal absolute position encodings from transformers (Vaswani et al., 2017). Given an electrode's position $(u, v)$ in the coordinate system, the embedding is defined as:

$$\begin{aligned} p_{u,v}(4k) &= \sin u\omega_k, \quad p_{u,v}(4k+1) = \cos u\omega_k, \\ p_{u,v}(4k+2) &= \sin v\omega_k, \quad p_{u,v}(4k+3) = \cos v\omega_k, \quad \omega_k = 1000^{-4k/d_e} \end{aligned} \tag{3}$$

where $k$ ranges from 0 to $\frac{d_e}{4}$, $d_e$ is the embedding dimension, and $\omega_k$ is the frequency term that ensures unique embeddings for up to 1000 positions by assigning different sinusoidal frequencies.

This approach allows signals from nearby electrode positions to receive similar embeddings. Specifically:

$$\text{Dist}((u_i, v_i), (u_j, v_j)) < \text{Dist}((u_i, v_i), (u_k, v_k)) \quad \implies \quad p_{u_i,v_i} \cdot p_{u_j,v_j} > p_{u_i,v_i} \cdot p_{u_k,v_k}$$

Where Dist() is the distance function in Cartesian coordinates, and $p_{u_i,v_i}, p_{u_j,v_j} \in \mathbb{R}^{d_e}$ represent the channel embeddings of the electrodes at positions $(u_i, v_i)$ and $(u_j, v_j)$, respectively. The equation demonstrates that electrodes closer in physical space (i.e., neighboring electrodes) will have higher similarity in their embeddings, as indicated by the dot product.

Fig. 2b shows the embedding similarity between electors $F4$ with all other electrodes. For example, as shown in Fig. 2b, signals from the $F4$ electrode will have embeddings more similar to those of $F2$ and $F6$

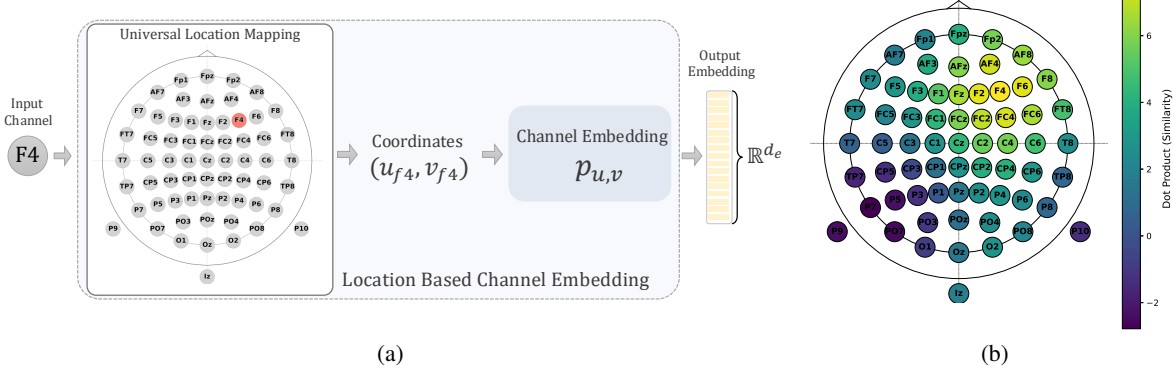

(a) (b)

Figure 2: (a) Location-based channel embedding using a universal scalp coordinate mapping to assign each channel Cartesian coordinates. (b) Embedding similarity of electrode F4 with all other channels, showing higher similarity to nearby electrodes (e.g., F2, F6) than distant ones (e.g., P7).

than to other electrodes (such as $P7$). As a result, when any of these electrodes are present in a dataset, the model can effectively utilize them. If an electrode is missing or too noisy to be useful, the model can reconstruct the signal primarily using data from adjacent electrodes. This approach mitigates the issue of missing channels across devices, as it maintains consistent channel relationships based on location, even if the channel labeling differs. Additionally, this encoding provides the model with spatial awareness, enabling it to recognize regional brain activity patterns, where neighboring areas exhibit similar behaviors while differing from other regions.

## 2.2 LATENT SPACE SELF-PREDICTION

EEG-X is pretrained with two objectives: latent space self-reconstruction and raw space reconstruction (Section 2.3). The goal of latent space self-reconstruction is to predict the representation of an EEG sample based on a masked view of the same input (Baevski et al., 2022; 2023; Assran et al., 2023; Mohammadi Foumani et al., 2024a). This method consists of masking, a teacher encoder, a student encoder, and a predictor. We adopt the Semantic Subsequence Preserving (SSP) method (Mohammadi Foumani et al., 2024a) to select masked tokens. Both teacher and student are 4-layer Transformers (Vaswani et al., 2017; Devlin et al., 2019); the teacher encodes unmasked inputs, while the student processes masked inputs, similar to MAE (He et al., 2022). A 2-layer cross-attention Transformer predicts masked token representations from the student to match the teacher's outputs. The training loss has two parts. First, the $L_2$ distance between predictions $\hat{Y}_i^B$ and targets $Y_i^B$:

$$\mathcal{L}_{\text{Align}}(\hat{Y}^B, Y^B) = \frac{1}{|B|} \sum_{j \in B} \|\hat{y}_j - y_j\|_2^2 \tag{4}$$

where $B$ is the set of masked patches. Teacher parameters are updated as an EMA of student parameters. Second, to ensure robust representations and prevent collapse, we add a regularization term ($\mathcal{L}_{\text{Reg}}$) (Bardes et al., 2021). This penalizes variance collapse and enforces covariance regularization across dimensions, encouraging the student to produce informative and diverse embeddings.

## 2.3 NOISE-AWARE RAW SPACE RECONSTRUCTION

While latent space reconstruction improves robustness to noise, we introduce a complementary raw space reconstruction loss to maximize the information content of representations from raw EEG. Previous models primarily reconstruct raw EEG signals, but this is challenging due to low signal-to-noise ratio, where artifacts often dominate the brain signal. EEG-X addresses this by reconstructing artifact-removed inputs, ensuring learned representations capture neural activity. Reconstruction loss choice is also critical: although MSE is the de facto objective in self-supervised reconstruction, it often overemphasizes high-amplitude components,

heavily penalizes large errors, overlooks frequency similarities, and fails to capture the shape of time-series patterns. EEG-X mitigates these issues with the **Di**ctionary **C**onvolution **T**ransformation (DiCT), which projects both artifact-removed inputs and reconstructions into a structured feature space, making the loss less sensitive to noise, balanced comparisons across frequency bands, and aware of shape similarity.

**Artifact Removal:** Our framework is agnostic to the choice of denoising method, allowing integration of any artifact-removal technique and ensuring flexibility across datasets and recording setups. In this work, we adopt ICA with ICLabel (Pion-Tonachini et al., 2019) as a strong starting point (see Appendix B.3 for a broader discussion of alternatives). After decomposing EEG into independent components, ICLabel uses a neural network classifier trained on large-scale expert-labeled datasets to assign probability scores across seven categories: brain activity, eye movements, muscle activity, heart signals, line noise, channel noise, and others. Components with high artifact probabilities are removed, and the remaining components are projected back to reconstruct a cleaner EEG signal. This provides a controlled reconstruction target, though alternative methods can be readily substituted without altering the framework.

**Dictionary Convolution Transformation (DiCT):** To improve reconstruction loss for EEG signals, we introduce *DiCT*, inspired by Hydra (Dempster et al., 2023). DiCT projects both the artifact-removed input $\mathbf{X}_{\text{clean}} \in \mathbb{R}^{C \times L}$ and the decoder output $\mathbf{X}_{\text{decoder}} \in \mathbb{R}^{C \times L}$ into a structured feature space before computing MSE. We use $G$ groups of random convolutional kernels, each with $K$ competing kernels. For each group $g$, convolutional responses are computed as

$$r_{g,k}(t) = (\mathbf{x} * \mathbf{w}_{g,k})(t), \tag{5}$$

where $*$ denotes 1D convolution. At each time step, kernels within a group *compete* and only the strongest (max or min) responses contribute to the dictionary representation. To capture information across multiple temporal scales, kernels are applied with dilations drawn from $d \in \{2^0, 2^1, 2^2, \dots\}$ subject to the constraint that the effective receptive field does not exceed the input length. The transformed representation of a signal $\mathbf{X}$ is denoted by $\mathcal{F}(\mathbf{X})$, formed by aggregating winning responses across groups and dilations. Reconstruction loss is then computed as

$$\mathcal{L}_{\text{Rec}}(\mathbf{X}_{\text{clean}}, \mathbf{X}_{\text{decoder}}) = \|\mathcal{F}(\mathbf{X}_{\text{clean}}) - \mathcal{F}(\mathbf{X}_{\text{decoder}})\|_2^2. \tag{6}$$

Compared to direct time-domain MSE, DiCT offers: (1) *Noise robustness* – random convolutions distribute error across kernels, reducing sensitivity to outlier artifacts; (2) *Frequency balance* – dilated kernels capture multi-scale temporal patterns, considering low- and high-frequency dynamics; (3) *Shape-awareness* – dictionary competition emphasizes structural similarity rather than pointwise amplitude matching (For a detailed demonstration of these properties, see Appendix B.1).

**Decoder and Training Objective:** We use a two-layer Transformer with a 1D transposed convolution (deconvolution) decoder, which takes as input the concatenated outputs of the predictor network and the student encoder to generate reconstructions. EEG-X is pre-trained with a combination of latent space prediction loss and the DiCT-enhanced raw space reconstruction loss:

$$\mathcal{L}_{\text{total}} = \mathcal{L}_{\text{Rec}} + \mathcal{L}_{\text{Align}} + \mathcal{L}_{\text{Reg}}. \tag{7}$$

## 3 EXPERIMENTAL RESULTS

We report the main experimental results of EEG-X, with additional setup details and supplementary analyses provided in Appendix C.

### 3.1 DATASETS AND PRE-PROCESSING

We evaluate EEG-X across seven diverse datasets from four EEG headsets, ranging from portable consumer-grade devices to clinical-grade systems, in both real-world and clinical settings. We use larger datasets for

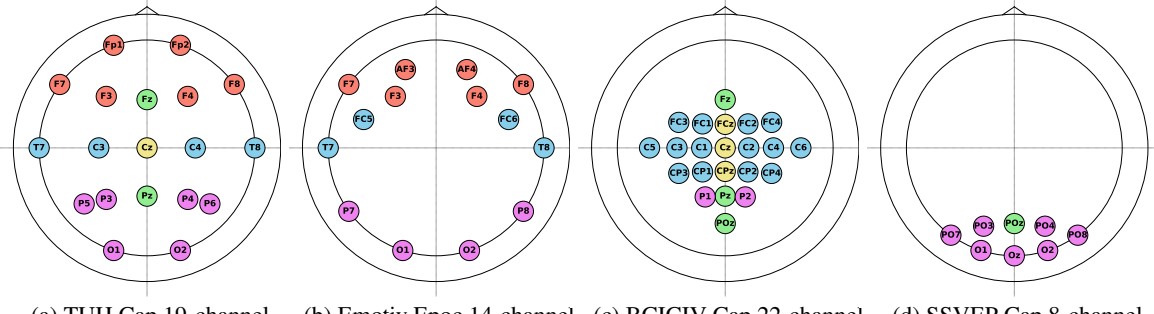

(a) TUH Cap 19-channel    (b) Emotiv Epoc 14-channel    (c) BCICIV Cap 22-channel    (d) SSVEP Cap 8-channel

Figure 3: Comparison of four different EEG headset configurations used for data collection

in-domain evaluation and smaller ones, recorded with different headsets and channel configurations (Fig. 3), for cross-domain evaluation. A summary of the datasets is provided in Table 1, with additional preprocessing details in Appendix C.2.

Table 1: Datasets for pre-training and downstream evaluation (headset configurations in Fig. 3).

| Category | Datasets | Channel | Length | #Samples | Task |
|---|---|---|---|---|---|
| **In-Domain** (Pre-training) | DREAMER | 14 | 2s | 77,910 | Emotion Detection |
| | STEW | 14 | 2s | 56,928 | Mental Workload Classification |
| | TUAB | 19 | 10s | 409,455 | Abnormal EEG Classification |
| | TUEV_binary | 19 | 2s | 98,684 | Epileptiform Classification |
| **Cross-Domain** (Downstream) | Crowdsourced | 14 | 2s | 12,296 | Eyes Open/Close Detection |
| | BCICIV_2A | 22 | 4s | 5,184 | Motor Imagery |
| | SSVEP | 8 | 2s | 960 | Steady-State Visual Potentials |

## 3.2 COMPARISON WITH BASELINE

We conducted comprehensive evaluations against six state-of-the-art EEG foundation models and representation learning: MAEEG (Chien et al., 2022), BioT (Yang et al., 2024), LabraM (Jiang et al., 2024), EEGPT (Wang et al., 2024), EEG2Rep (Mohammadi Foumani et al., 2024a), and CBraMod (Wang et al., 2025), using publicly available code to ensure fair comparisons. We report **Accuracy** (balanced), defined as the average recall across classes (*Acc* for binary and *B-Acc* for multi-class tasks). For binary classification, we also report **AUROC** (Area Under the ROC Curve) to summarize performance across thresholds. For multi-class tasks, we use the **Weighted F1** score, the class-weighted average of F1 scores proportional to sample counts (see Appendix C.1 for experimental details).

Table 2: Average classification accuracy over five runs in the in-domain setting across four EEG tasks

| Model | STEW | | DREAMER | | TUAB | | TUEV_binary | |
|---|---|---|---|---|---|---|---|---|
| | Acc | AUROC | Acc | AUROC | Acc | AUROC | Acc | AUROC |
| MAEEG | 75.98 $\pm_{2.87}$ | 81.73 $\pm_{2.95}$ | 52.36 $\pm_{3.12}$ | 54.45 $\pm_{2.98}$ | 77.56 $\pm_{1.45}$ | 86.56 $\pm_{1.67}$ | 85.75 $\pm_{2.03}$ | 91.36 $\pm_{1.89}$ |
| BioT | 75.79 $\pm_{1.12}$ | 86.48 $\pm_{1.45}$ | 52.85 $\pm_{1.34}$ | 54.19 $\pm_{1.23}$ | 79.21 $\pm_{1.56}$ | 87.42 $\pm_{1.78}$ | 86.28 $\pm_{1.43}$ | 93.06 $\pm_{1.67}$ |
| LaBraM | 75.26 $\pm_{1.35}$ | 84.37 $\pm_{1.67}$ | 55.85 $\pm_{1.45}$ | 55.34 $\pm_{1.23}$ | 80.61 $\pm_{1.54}$ | 88.77 $\pm_{1.76}$ | 85.43 $\pm_{1.48}$ | 90.10 $\pm_{1.59}$ |
| EEGPT | 77.65 $\pm_{2.95}$ | 85.98 $\pm_{3.01}$ | 54.11 $\pm_{3.12}$ | 54.91 $\pm_{2.87}$ | 78.91 $\pm_{1.49}$ | 87.16 $\pm_{1.85}$ | 84.05 $\pm_{2.03}$ | 90.26 $\pm_{1.78}$ |
| EEG2Rep | 76.71 $\pm_{1.23}$ | 85.89 $\pm_{1.45}$ | 57.09 $\pm_{1.32}$ | 56.35 $\pm_{1.21}$ | 80.21 $\pm_{1.54}$ | 88.43 $\pm_{1.67}$ | 89.83 $\pm_{1.38}$ | 95.57 $\pm_{1.47}$ |
| CBraMod | 75.89 $\pm_{1.43}$ | 84.89 $\pm_{1.37}$ | 56.09 $\pm_{1.09}$ | 55.56 $\pm_{1.21}$ | 80.05 $\pm_{1.4}$ | 88.63 $\pm_{1.7}$ | 87.35 $\pm_{1.42}$ | 94.71 $\pm_{1.25}$ |
| **EEG-X** | **80.32** $\pm_{1.21}$ | **88.37** $\pm_{1.43}$ | **58.14** $\pm_{1.35}$ | **57.64** $\pm_{1.29}$ | **81.02** $\pm_{1.48}$ | **88.97** $\pm_{1.62}$ | **91.15** $\pm_{1.41}$ | **96.18** $\pm_{1.39}$ |

Table 3: Cross-domain performance comparison of models pre-trained on TUAB dataset.

| Model | Crowdsourced | | BCICIV_2A | | SSVEP | |
|---|---|---|---|---|---|---|
| | Acc | AUROC | B-Acc | Weighted F1 | B-Acc | Weighted F1 |
| BioT | 77.76 $\pm_{1.12}$ | 83.62 $\pm_{1.34}$ | 27.26 $\pm_{1.05}$ | 25.26 $\pm_{1.23}$ | 42.12 $\pm_{1.41}$ | 40.17 $\pm_{1.36}$ |
| LaBraM | 77.28 $\pm_{1.25}$ | 89.83 $\pm_{1.47}$ | 32.44 $\pm_{1.13}$ | 27.37 $\pm_{1.21}$ | 38.89 $\pm_{1.28}$ | 37.31 $\pm_{1.19}$ |
| EEGPT | 79.55 $\pm_{1.36}$ | 87.79 $\pm_{1.52}$ | 30.07 $\pm_{1.17}$ | 24.68 $\pm_{1.08}$ | 24.14 $\pm_{1.11}$ | 23.50 $\pm_{1.07}$ |
| CBraMod | 78.41 $\pm_{1.31}$ | 88.73 $\pm_{1.45}$ | 30.07 $\pm_{1.12}$ | 24.68 $\pm_{1.09}$ | 41.44 $\pm_{1.34}$ | 39.69 $\pm_{1.25}$ |
| **EEG-X** | **86.89 $\pm_{1.28}$** | **93.51 $\pm_{1.41}$** | **37.58 $\pm_{1.19}$** | **35.32 $\pm_{1.15}$** | **57.17 $\pm_{1.42}$** | **55.05 $\pm_{1.33}$** |

**In-domain evaluation:** Table 2 reports average classification accuracy over five runs, compared with other state-of-the-art methods. **Bold** indicates the highest accuracy for each dataset, while underlined values denote the second-best, consistent across all tables in this paper. In the in-domain evaluation, the model is pre-trained and fine-tuned on the same dataset, ensuring that EEG-X is evaluated within a consistent recording setup for each task. The results show that EEG-X achieves the highest average performance across all EEG tasks. Latent space reconstruction models like EEG2Rep outperform raw space reconstruction models such as MAEEG and EEGPT on noisy datasets like TUAB and TUEV_binary. EEG-X surpasses both by using artifact removal and the DiCT-enhanced reconstruction loss instead of raw signals, enabling it to learn noise-robust representations while retaining maximal information from the input, leading to strong performance on these challenging datasets.

**Cross-domain evaluation as a foundation model:** Table 3 presents the average classification accuracy of EEG-X, highlighting its role as a *foundation model* for EEG. It is compared to models using a channel-wise tokenizer, which allows them to handle cross-domain scenarios where the number of channels differs between pre-training and downstream tasks (note that MAEEG and EEG2Rep are excluded as they use convolutional tokenizers). In this setup, all models were pre-trained solely on the TUAB dataset, with the pre-trained tokenizer and student encoder frozen, and linear regression applied to the learned representations (outputs from the student encoder).

As shown in this table, EEG-X, acting as a foundation model, significantly outperforms other models across all tasks. The pre-training dataset (19 TUH configurations) and downstream tasks (14-, 22-, and 8-channel setups) share few channels but exhibit higher channel similarity. By using location-based channel embeddings, EEG-X effectively preserves spatial relationships among neighboring channels, even across different devices. This enables strong generalization when the pre-trained foundation model is applied to downstream tasks with varying devices, channel counts, and configurations.

### 3.3 Ablation study

**Channel Embedding**: We evaluate the impact of channel embeddings on model performance in both in-domain and cross-domain scenarios, comparing three approaches: No Channel Embedding (✗ CE), learnable Channel Embedding (CE), and our novel Location-based Channel Embedding (EEG-X). As shown in Table 4, models with learnable and location-based embeddings consistently outperform those without. Notably, our approach (EEG-X), which incorporates spatial information to make the model location-aware, achieves the largest gains. This is particularly evident in cross-domain settings, where Loc-CE preserves spatial relationships among neighboring channels, allowing the model to capture regional brain activity patterns and maintain consistent channel relationships across datasets with different channel counts and configurations.

**DiCT**: We further assess the effect of introducing the Dictionary Convolution Transformation (DiCT) on the reconstruction loss. As shown in Table 4, this leads to more stable and discriminative representations by emphasizing higher-level temporal and spectral patterns. While the improvements are moderate in in-domain datasets, DiCT demonstrates clearer benefits in cross-domain transfer, where structured feature alignment helps the model generalize across diverse recording setups and subject variability.

**Artifact Removal Raw Space**: We examined the impact of artifact-removed raw space reconstruction by comparing the reconstruction of raw EEG signals with our method, which reconstructs the ICA-cleaned version. As shown in Table 4, reconstructing ICA-cleaned signals outperforms the same model using raw reconstruction loss, indicating that denoising EEG data enhances the semantic quality of learned representations by reducing noise from the raw input. Additionally, incorporating raw space reconstruction alongside latent space reconstruction leads to performance gains in less noisy datasets like STEW.

Table 4: Ablation study of EEG-X components

| Category | Dataset | ✗CE | ✓CE | ✗ DiCT | ✗$\mathcal{L}_{rec}$ | $\mathcal{L}_{rec\,(Raw)}$ | EEG-X |
|---|---|---|---|---|---|---|---|
| **In-Domain** | STEW | 79.44 | 80.07 | 76.91 | 75.92 | 77.90 | **80.32** |
| | DREAMER | 56.17 | 57.12 | 54.36 | 53.12 | 55.41 | **58.14** |
| | TUAB | 79.94 | 80.94 | 78.88 | 78.15 | 79.41 | **81.02** |
| | TUEV | 90.46 | 90.14 | 88.01 | 87.91 | 88.22 | **91.15** |
| | **Average** | (↓ 1.16) | (↓ 0.43) | (↓ 3.10) | (↓ 3.88) | (↓ 2.42) | 77.66 |
| **Cross-Domain** | Crowdsourced | 81.64 | 81.65 | 84.12 | 82.22 | 85.34 | **86.89** |
| | BCICIV_2a | 33.55 | 33.25 | 34.57 | 33.21 | 35.31 | **37.58** |
| | SSVEP | 40.02 | 41.2 | 51.55 | 45.68 | 52.22 | **57.17** |
| | **Average** | (↓ 8.81) | (↓ 8.52) | (↓ 3.80) | (↓ 6.85) | (↓ 2.93) | 60.55 |

**ICA pre-processing vs ICA_Loss:** A key question is: *Why not use artifact removal (ICA) directly as a preprocessing step rather than as a loss?* Our experiments show that while ICA effectively removes artifacts, it can also result in the loss of essential brain signal components, which can negatively affect downstream tasks. As shown in Table 5, we evaluate EEG-X (trained supervisedly for 100 epochs) on both raw data and ICA-preprocessed data to assess the discriminative power after preprocessing. The results demonstrate that ICA preprocessing is not always beneficial, as it can remove important brain signal components, leading to greater signal loss compared to using raw data. Therefore, integrating artifact removal within our model allows for a more controlled and optimized reconstruction process.

Table 5: Performance comparison of EEG-X on raw vs. ICA-preprocessed data across various datasets, highlighting the impact of using ICA either as a preprocessing step or as a loss function.

| Dataset | STEW | DREAMER | TUAB | TUEV |
|---|---|---|---|---|
| Raw | **75.12** | 55.69 | 78.12 | **88.24** |
| ICA Pre-processed | 70.68 | **56.24** | **78.25** | 83.36 |

## 4 CONCLUSION

In this paper, we introduced *EEG-X*, a device-agnostic and noise-robust foundation model for EEG representation learning. EEG-X tackles two fundamental challenges in EEG analysis: variability across datasets and the low signal-to-noise ratio of EEG signals. By integrating a noise-aware dual reconstruction strategy—reconstructing denoised signals in both raw and latent spaces—our framework ensures that the learned representations emphasize neural activity rather than noise. Furthermore, the proposed location-based channel embedding captures spatial relationships among electrodes, enabling effective generalization across datasets with different electrode layouts and recording configurations. The introduction of the Dictionary Convolution Transformation (DiCT) further enhances reconstruction by projecting signals into a structured feature space, improving robustness and capturing frequency- and shape-aware similarities. Extensive experiments across diverse datasets demonstrate that EEG-X not only outperforms state-of-the-art methods in in-domain and cross-domain settings but also establishes a strong foundation model for EEG analytics.

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

## A    RELATED WORK

A line of work tackles the variability in EEG data caused by differences in recording devices and configurations, particularly the difference in the number of channels and input lengths. BIOT (Yang et al., 2024) tackles this challenge by tokenizing biosignals into a "sentence-like" structure, segmenting EEG time series in each channel into fixed-length tokens, and adding temporal and channel embedding. Similarly, EEGformer (Wan et al., 2023), LaBraM (Jiang et al., 2024), and EEGPT (Wang et al., 2024) tokenize EEG signals on a channel-wise basis and process them using transformer-based architectures. LaBraM uses learnable embeddings for both channel and temporal positions, while EEGPT incorporates rotary position encodings for temporal representation, alongside learnable channel embeddings. However, these learnable channel embeddings often lack spatial awareness, as they do not reflect the placement of electrodes on the scalp —critical for capturing meaningful EEG patterns. Moreover, the same electrode might not exist in different devices, making these embeddings impractical for handling channel mismatches.

A parallel line of work has explored self-prediction approaches, such as Masked Autoencoders (He et al., 2022), to adapt pre-trained models for EEG (Chien et al., 2022; Kostas et al., 2021; Wang et al., 2024; Mohammadi Foumani et al., 2024a; Wang et al., 2025). BENDER (Kostas et al., 2021), MAEEG (Chien et al., 2022), and CBraMod (Wang et al., 2025) use masking and reconstruction as pretext tasks for self-supervised learning, directly reconstructing masked segments in the raw signal space. However, raw space reconstruction is highly sensitive to noise, often leading to poor representation quality. Alternatively, LaBraM (Jiang et al., 2024) uses a neural tokenizer that transforms raw EEG signals into vocabulary-based tokens, predicting tokens instead of reconstructing raw signals. However, since these models focus on raw space for representation learning, they suffer from a low signal-to-noise ratio, with data often buried under external artifacts and non-brain sources. EEG2Rep (Mohammadi Foumani et al., 2024a) improves robustness against noise by introducing latent space reconstruction instead of raw space, reducing sensitivity to noise and encouraging the model toward semantically rich representations by capturing high-level features. However, it remains limited in handling datasets with varying number of channels and recording lengths.

## B    EXTENDED METHODOLOGY

### B.1    DEMONSTRATING FREQUENCY- AND SHAPE-AWARE RECONSTRUCTION WITH DiCT

To illustrate the limitations of direct time-domain MSE and the benefits of the proposed Dictionary Convolution Transformation (DiCT), we conducted a controlled synthetic experiment.

**Setup.** We constructed a composite source signal by summing three sine waves (Fig. 4, bottom):

- Low frequency (2 Hz) with high amplitude (5.0)
- Mid frequency (20 Hz) with moderate amplitude (1.0)
- High frequency (100 Hz) with low amplitude (0.5)

This design reflects real EEG signals, where most energy is concentrated in low frequencies but informative mid- and high-frequency components are present with smaller amplitudes. The individual components are shown in Fig. 4 (top three panels).

**Targets.** We generated three reconstructed signals with varying distortions from a combined input signal (Fig. 5):

1. *Low-frequency dominant reconstruction*, preserving mainly the strong low-frequency component.
2. *High-frequency dominant reconstruction*, emphasizing mid/high components while attenuating low-frequency energy.

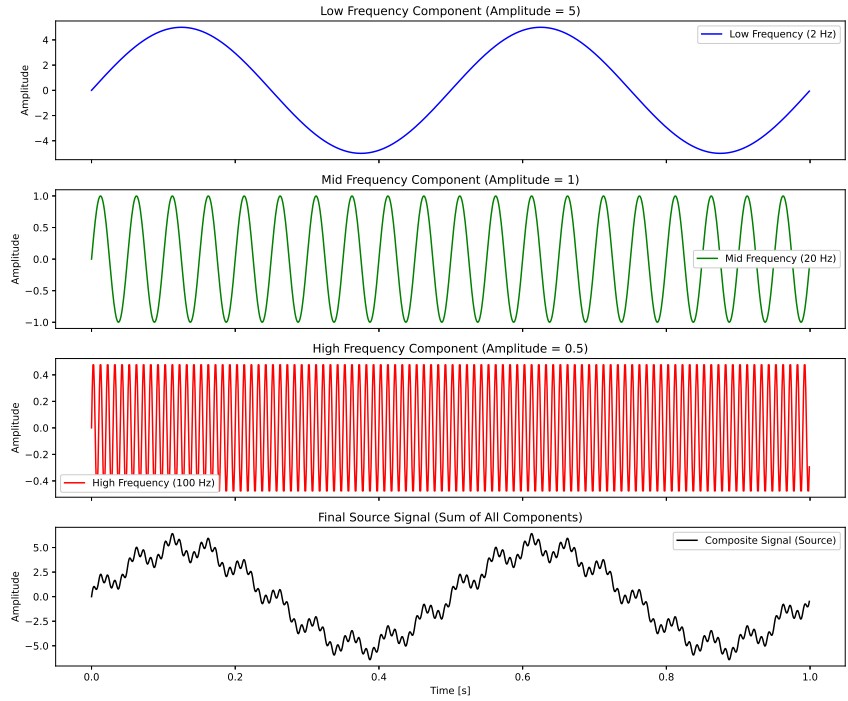

Figure 4: Composite source signal used in the synthetic experiment, formed by summing three sine waves: low frequency (2 Hz, amplitude 5.0), mid frequency (20 Hz, amplitude 1.0), and high frequency (100 Hz, amplitude 0.5).

3. *High-frequency dominant + phase-shift reconstruction*, introducing both spectral imbalance and temporal misalignment through phase shifts.

As shown in Table 6, direct time-domain MSE is strongly biased toward amplitude. The low-frequency reconstruction—though missing most mid/high content—yields a very small error (MSE = 0.58), whereas reconstructions with more balanced frequency content but less amplitude alignment are heavily penalized (MSE = 6.14 and 13.88). This demonstrates MSE's tendency to reward amplitude matching while largely ignoring spectral and temporal structure. Applying DiCT before error computation mitigates this bias: random convolutional projections redistribute the influence across frequencies, penalizing the low-frequency-only reconstruction more heavily (MSE = 9.26) and giving high-frequency-rich reconstructions more comparable errors (MSE = 6.74 and 7.25).

These results highlight several key points. First, random convolutions improve noise robustness by distributing errors across kernels, reducing over-penalization by local artifacts. Second, the use of dilated kernels captures both low- and high-frequency dynamics, ensuring that informative fast components are no longer ignored. Finally, the phase-shifted reconstruction demonstrates a critical limitation of direct MSE: signals with identical frequency spectra but temporal misalignment are scored as drastically different (MSE = 13.88). DiCT, however, compares representations in a convolutional dictionary space where temporal shape similarity is preserved, reducing the error to 7.25. This shows DiCT's ability to recognize structural alignment even under shifts—a property especially important for EEG signals, where phase variability across trials and subjects is common.

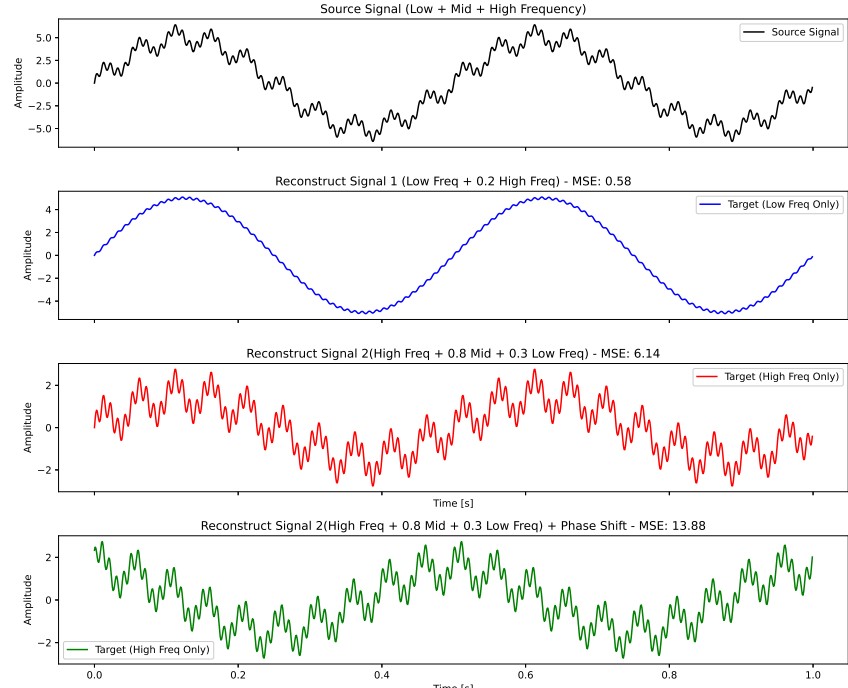

Figure 5: Three reconstructed signals generated from a combined input signal, illustrating different distortions: (1) low-frequency dominant, (2) high-frequency dominant, and (3) high-frequency dominant with phase shifts. These examples highlight the bias of direct MSE toward amplitude and the mitigating effect of the DiCT transformation (see Table 6 for quantitative results).

| Reconstruction Type | MSE (Direct Time) | MSE with DiCT |
|---|---|---|
| Low-frequency only (2 Hz + 0.2 High) | 0.58 | 9.26 |
| High-frequency + mid + partial low | 6.14 | 6.74 |
| High-frequency + mid + low + phase shift | 13.88 | 7.25 |

Table 6: Comparison of reconstruction error measured with direct MSE versus DiCT-enhanced MSE on synthetic signals.

## B.2 UNIVERSAL LOCATION MAPPING

Figure 6 shows a dense scalp mesh including 348 channels, covering virtually all electrode positions from the 10-05, 10-10, and 10-20 EEG systems. This mesh enables universal location-based channel embeddings: even if electrode names differ across devices, each sensor can be mapped to the nearest mesh point. Since typical EEG sensors are larger than 5 mm in diameter, any sensor smaller than 10 mm placed anywhere on the scalp will intersect with at least one mesh point, ensuring consistent spatial representation across devices and subjects.

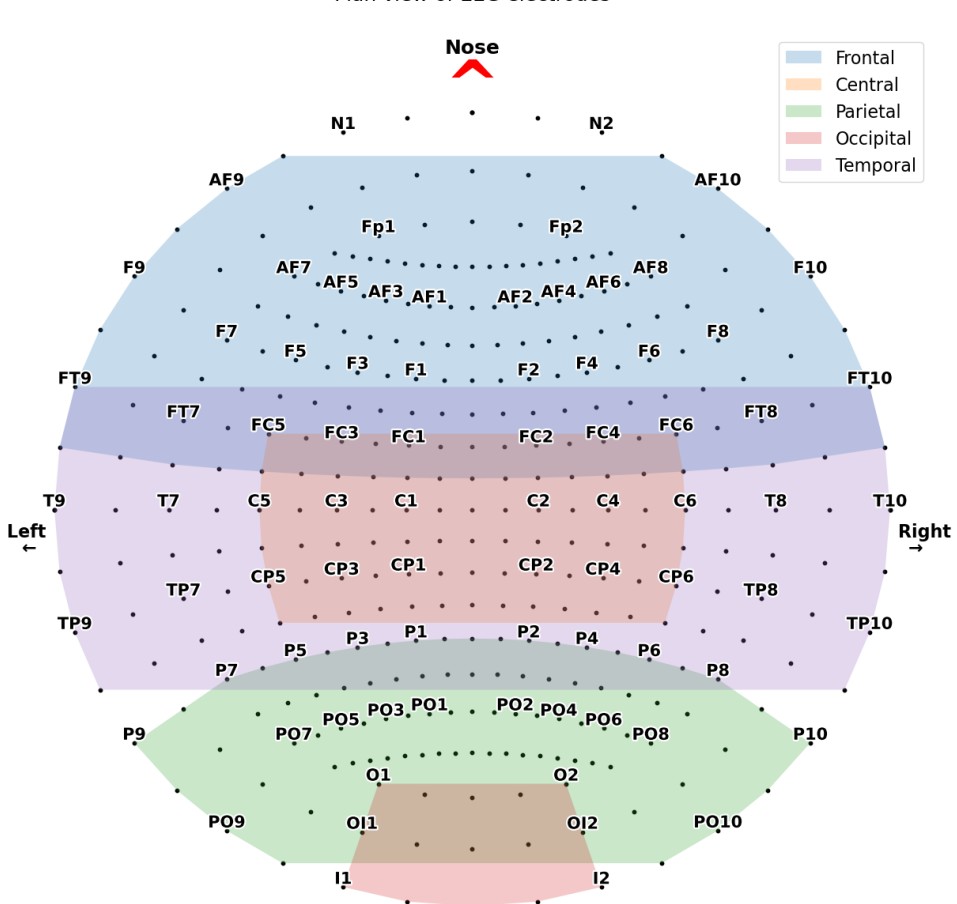

Figure 6: Dense scalp electrode mesh combining positions from the 10-05, 10-10, and 10-20 systems, used as a universal location mapping.

## B.3 ARTIFACT REMOVAL DETAILS

We chose ICA-based denoising because it effectively isolates spatially and temporally independent artifacts. By separating these sources, ICA provides a cleaner supervisory target for EEG reconstruction without manual labeling or handcrafted filters. It is widely regarded as a gold standard for isolating artifacts such as muscle activity, eye blinks, cardiac signals, respiration, motion artifacts, and power line noise, which can be 5–50 times larger than typical brain signals.

We incorporate ICLabel Pion-Tonachini et al. (2019), a neural network classifier trained on a large-scale EEG dataset, to automatically label the independent components extracted via ICA. ICLabel assigns probability scores to each component across seven categories: *brain activity, eye movements, muscle activity, heart signals, line noise, channel noise,* and *others*. Components with high artifact probabilities are removed, and the remaining components are projected back to the raw space to reconstruct a clean EEG signal.

The ICA-based artifact removal workflow is as follows:

1. **ICA Decomposition:** EEG data is decomposed into independent components using ICA.

2. **IC Classification:** ICLabel assigns probability scores to each component.

3. **Artifact Identification:** Components classified as artifacts based on threshold values are removed.

4. **Reconstruction:** Remaining components are projected back to reconstruct a cleaner EEG signal.

Alternative artifact removal methods include:

- **rASR (Riemannian Artifact Subspace Reconstruction):** Requires clean baseline data to train a joint covariance matrix; effective for real-time denoising.
- **Wavelet Denoising:** Particularly useful for high-frequency noise due to its time-frequency localization, but lacks ICA's spatial decomposition capabilities.
- **Regression-based Methods:** Use auxiliary channels (e.g., EOG, EMG) to remove artifacts, but such channels are not always available.
- **Spatial Filters (e.g., CSP):** Effective for task-specific artifact removal but may not generalize across different EEG paradigms.

Despite ICA's assumptions (e.g., statistical independence of sources), it offers a strong trade-off between generality and effectiveness for large-scale, domain-agnostic EEG analysis. Our framework is compatible with alternative denoising methods, but ICA with ICLabel provides a reliable starting point.

## C  ADDITIONAL EXPERIMENTAL DETAILS AND RESULTS

### C.1  EXPERIMENTAL SETUP

In our experiments, EEG-X was trained with a batch size of 256 using the Adam optimization algorithm (Kingma and Ba, 2014). To prevent overfitting, we implemented early stopping based on the validation loss. The model was pre-trained for 300 epochs. For fine-tuning, the tokenizer, context encoder, and classification head were trained for 10 epochs in a fully supervised manner using cross-entropy loss. In the cross-domain setting, the tokenizer and context encoder were frozen, and logistic regression was applied to the learned representations.

Similar to the default transformer block(Vaswani et al., 2017), we used eight attention heads in both the context and target encoders to capture diverse features from the EEG data. The transformer encoding dimension was set to $d_e = 16$, and the feed-forward network (FFN) in the transformer block expanded the input size by a factor of 4 before projecting it back to its original size. The learning rate was initialized at $1 \times 10^{-3}$ and adjusted over time using a cosine learning rate scheduler (Baevski et al., 2022). EEG signals were segmented using a patch size of 128 (1 second) with an overlap of 32 time steps, ensuring a balance between capturing temporal dynamics and maintaining computational efficiency.

For the Dictionary Convolution Transformation (DiCT), we followed the Hydra setup (Dempster et al., 2023) and employed $G = 32$ groups of random convolutional kernels, each containing $K = 8$ competing kernels. This configuration balances efficiency and representational diversity, while ensuring robustness across frequency scales.

### C.2  DATASET OVERVIEW AND PROCESSING

For the Temple University Hospital (TUH) datasets, we use the default train and test splits. For the remaining datasets, we randomly select 20% of the data subject-wise, ensuring that the same subject is not present in both the training and test sets.

### C.2.1 EMOTIV

We applied a bandpass filter to all Emotiv datasets and used 13 components for ICA-based artifact removal. We then discarded all components except those representing brain activity and other relevant signals. The data was then segmented into windows of 256 time steps, with each window corresponding to 2 seconds of recording.

**DREAMER** DREAMER is a multimodal dataset containing electroencephalogram (EEG) and electro-cardiogram (ECG) signals recorded during affect elicitation using audio-visual stimuli (Katsigiannis and Ramzan, 2017), captured with a 14-channel Emotiv EPOC headset. The dataset includes data from 23 participants, along with their self-assessments of affective states (valence, arousal, and dominance) after each stimulus. For our classification task, we focus specifically on the arousal labels. The DREAMER dataset can be accessed here[1]. We use the raw EEG data, applying low-pass and high-pass filters, and then performing ICA with 13 components for artifact removal[2]. It's important to note that for our analysis, we exclude the ECG signals and only focus on the EEG data.

**Crowdsourced** Crowdsourced EEG data was collected while participants were engaged in a resting state task, involving periods with eyes open and eyes closed, each lasting 2 minutes. Out of 60 participants, only 13 successfully completed both conditions using 14-channel EPOC+, EPOC X, and EPOC devices. The data was initially recorded at 2048 Hz and later downsampled to 128 Hz. Raw EEG data for these 13 participants, along with preprocessing, analysis, and visualization scripts, are openly available on the Open Science Framework (OSF)[3].

**Simultaneous Task EEG Workload (STEW)** STEW dataset comprises raw EEG recordings from 48 participants using a 14-channel Emotiv EPOC headset involved in a multitasking workload experiment utilizing the SIMKAP multitasking test (Lim et al., 2018). Additionally, the subjects' baseline brain activity at rest was recorded before the test. The data was captured using the Emotiv Epoc device with a sampling frequency of 128Hz and 14 channels, resulting in 2.5 minutes of EEG recording for each case. Participants were instructed to assess their perceived mental workload after each stage using a rating scale ranging from 1 to 9, and these ratings are available in a separate file. Moreover, this dataset includes binary class labels, considering a workload rating of more than 4 as high and otherwise as low. We utilize these labels for our specific problem. STEW can be accessed upon request through the IEEE DataPort[4].

### C.2.2 TEMPLE UNIVERSITY HOSPITAL (TUH) EEG CORPUS

**TUH Abnormal EEG Corpus (TUAB)**
The TUH Abnormal EEG Corpus (TUAB) is a subset of the Temple University Hospital (TUH) EEG Corpus, which is one of the largest publicly available collections of clinical EEG data. The TUAB specifically focuses on EEG recordings labeled as abnormal, making it a valuable resource for studies on neurological disorders, brain function anomalies, and the development of diagnostic tools (Lopez et al., 2015).

**TUH EEG Events (TUEV)**
The TUH EEG Events Corpus (TUEV) contains annotations of EEG segments classified into six different categories: spike and sharp wave, generalized periodic epileptiform discharges, periodic lateralized epileptiform discharges, eye movement, artifact, and background (Harati et al., 2015).

**Acquisition and Preprocessing**
The TUH Abnormal EEG Corpus (TUAB) (Lopez et al., 2015) and TUH EEG Events (TUEV) (Harati

---

[1] https://zenodo.org/records/546113
[2] https://torcheeg.readthedocs.io/en/v1.1.0/torcheeg.datasets.html
[3] https://osf.io/9bvgh
[4] https://ieee-dataport.org/open-access/stew-simultaneous-task-eeg-workload-dataset

et al., 2015) can be accessed upon request through the Temple University Electroencephalography (EEG) Resources[5]. We processed both datasets to adhere to the 19-channel EEG montage following the 10-20 international system. The specific electrode configuration is as follows:

```
TUH_19_electrodes = ['fp1', 'fp2', 'f3', 'f4', 'c3', 'c4', 'p3', 'p4',
'o1', 'o2', 'f7', 'f8', 't7', 't8', 'p6', 'p5', 'fz', 'cz', 'pz']
```

For artifact removal, we applied ICA using 18 components for cleaning the data.

### C.2.3 BCICIV-2A: MOTOR IMAGERY TASK:

The BCIC-IV-2a Brunner et al. (2008) dataset was recorded using a 22-channel EEG headset (Fig.3c). The data were downsampled from 200 Hz to 128 Hz and consist of EEG recordings from 9 subjects performing four motor imagery tasks: left hand, right hand, feet, and tongue. Each subject completed two sessions on separate days, with 288 trials per session. This dataset was selected for its minimal electrode overlap with Emotiv and TUH devices (as shown in Fig. 3), providing a challenging cross-domain evaluation.

### C.2.4 ONLINE SSVEP-BASED BCI USING RIEMANNIAN GEOMETRY:

The dataset comprises 8-channel EEG recordings from 12 subjects (ages 20–28) performing SSVEP-based tasks (Kalunga et al., 2016). Each trial lasts 5 seconds, initiated by an auditory cue indicating which LED to focus on, or a fixation point for the reject class. The data were resampled to 128 Hz and segmented into 2-second windows. The primary task involves classifying the subject's focus on one of three LED groups or the reject class, utilizing Riemannian geometry for classification. The dataset is publicly available for research purposes.

### C.3 VISUALIZATION

This section visualizes the evolution of learned representations during pre-training using t-SNE plots at various epoch. As shown in Fig. 7 In the early stages of training (e.g., epoch 5 and 10 Fig.7a and 7b, respectively), the embeddings from the two classes are highly entangled, with little distinction between them, showing that the model is just beginning to understand the data's structure. As training progresses, class separation improves, with noticeable distinction by epoch 100 (Fig.7c) and clear separation by epoch 300 (Fig.7e). Despite being unsupervised, the model leverages the STEW dataset's inherent structure to enhance class separation. The model's accuracy also increases from 65.73% at epoch 5 to 78.81% at epoch 300, highlighting the benefits of self-supervised training in learning meaningful representations.

---

[5]https://isip.piconepress.com/projects/tuh_eeg/html/downloads.shtml

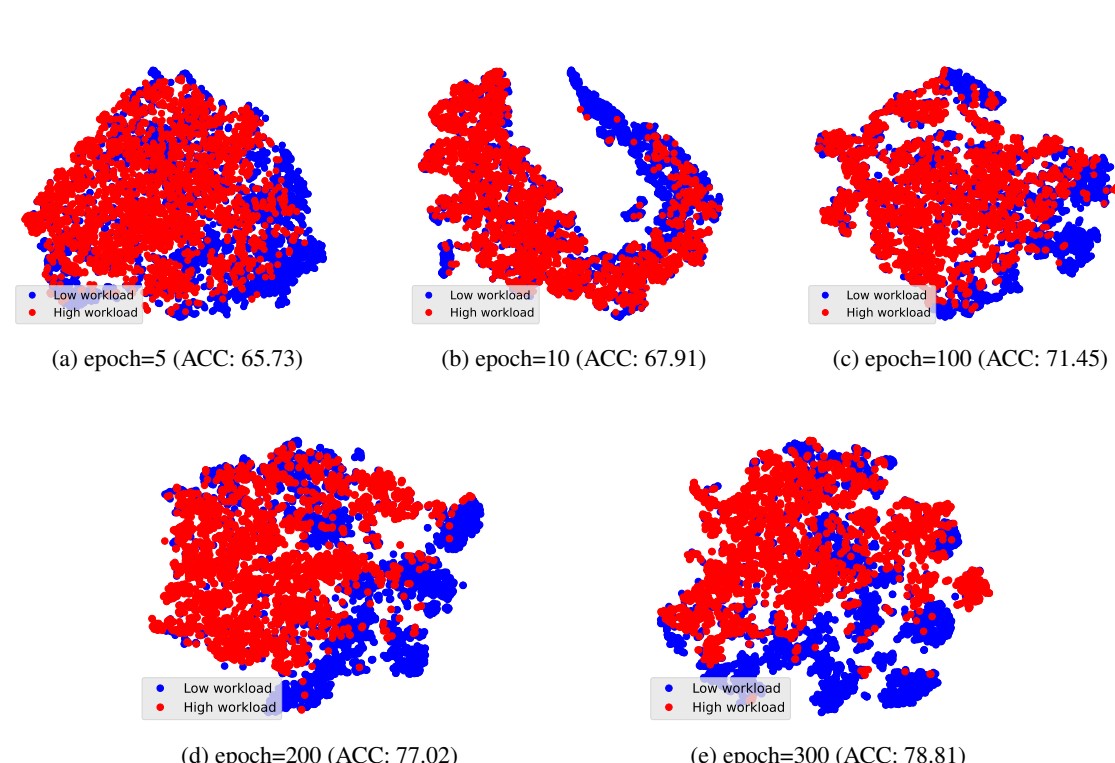

(a) epoch=5 (ACC: 65.73)  (b) epoch=10 (ACC: 67.91)  (c) epoch=100 (ACC: 71.45)

(d) epoch=200 (ACC: 77.02)  (e) epoch=300 (ACC: 78.81)

Figure 7: 2D t-SNE visualization of the learned representation on the STEW dataset at different epochs during pre-training. At epoch 5, the classes are highly entangled, with separation increasing over time, becoming distinct by epoch 300. The accuracy improves from 65.73% at epoch 5 to 78.81% at epoch 300.

