# OpenReview forum: "EEG-X: Device-Agnostic and Noise-Robust Foundation Model for EEG"
_ICLR.cc/2026/Conference — Submitted to ICLR 2026_

### Official Review · Reviewer_UvbJ · 2025-10-27

**Soundness:** 2
**Presentation:** 3
**Contribution:** 1
**Rating:** 2
**Confidence:** 5

**Summary:**

The paper introduces EEG-X, a device-agnostic and noise-robust foundation model for EEG representation learning. The model addresses two primary challenges in EEG analysis: the variability across different EEG devices and configurations, and the inherent low signal-to-noise ratio (SNR) of EEG data. EEG-X incorporates a location-based channel embedding, which encodes the spatial relationships between electrodes to ensure robustness across different devices. Additionally, it employs a noise-aware masking-reconstruction strategy, working in both raw and latent spaces to enhance the robustness of the learned representations. A key component of EEG-X is the DiCT layer, which helps capture frequency- and shape-aware similarities, improving noise robustness during signal reconstruction. The authors demonstrate the effectiveness of EEG-X across several datasets and tasks, showcasing its superior performance over current methods.

**Strengths:**

The paper is well-written and easy to follow. The structure is logical, and the explanations of the model architecture and methods are concise and clear, making it accessible to readers from different fields. The paper compares EEG-X with several recent state-of-the-art models in the experiments.

**Weaknesses:**

The work lacks significant novelty, for the following reasons:

1. The model architecture is quite similar to that of EEG2Rep [1], as it also uses two encoders combined with EMA for learning. The masked strategy used in the MAE is the same as in existing models.

2. In the latent reconstruction component, the structure consists of an encoder and predictor, which mirrors the approach in EEG2Rep.

3. Furthermore, the idea of latent reconstruction itself is not new, as models like EEGPT [2] have already explored this strategy.

4. In the denoising reconstruction part, this approach has also been explored in related works, such as DMAE-EEG [3], which also uses denoised signals as supervision.

5. Although the paper proposes a location-based channel embedding to handle varying electrode configurations, this method essentially uses a 2D position encoding similar to what is done in image processing. Mapping EEG electrode positions to a 2D spatial grid is not a novel approach. Additionally, other studies, such as BrainGPT [4], have already provided innovative solutions to address the issue of varying electrode configurations. Hence, the contribution in this regard is not groundbreaking.

Overall, the individual components proposed in this work do not show strong novelty or significant contribution. Some of the methods appear to be patchwork or combinations of existing techniques rather than offering a fresh perspective. The contributions in terms of methodology are not sufficiently robust.

The experiments in this work are rather thin, reflected in:

1. The experiments conducted in this paper are relatively simple. There is a lack of parameter experiments and comparisons with different position encoding methods.

2. The number of datasets and tasks considered is limited, with the paper focusing mainly on datasets with smaller electrode counts (8, 14, 19, 22), which are not very representative of the wide variety of EEG data. The inclusion of more diverse datasets and tasks, particularly those with more electrodes, would have enhanced the paper's scope and relevance.

3. The experimental setup lacks a comprehensive evaluation of the model's performance across a broader range of scenarios.

While the work is well-written, the overall approach is fairly basic. The methods do not provide sufficient novel insights, and the experimental results lack the complexity and depth needed to justify the model's contribution to the field. The paper presents a somewhat incremental approach rather than offering a breakthrough in EEG analysis.

[1] Mohammadi Foumani, Navid, et al. "Eeg2rep: enhancing self-supervised eeg representation through informative masked inputs." Proceedings of the 30th ACM SIGKDD Conference on Knowledge Discovery and Data Mining. 2024.

[2] Wang, Guangyu, et al. "Eegpt: Pretrained transformer for universal and reliable representation of eeg signals." Advances in Neural Information Processing Systems 37 (2024): 39249-39280.

[3] Zhang, Yifan, et al. "DMAE-EEG: A Pretraining Framework for EEG Spatiotemporal Representation Learning." IEEE Transactions on Neural Networks and Learning Systems (2025).

[4] Yue, Tongtian, et al. "Eegpt: Unleashing the potential of eeg generalist foundation model by autoregressive pre-training." arXiv preprint arXiv:2410.19779 (2024).

**Questions:**

1. Regarding the novelty issue, please refer to the detailed content in the Weakness section.

2. The claim that DiCT provides noise robustness, frequency balance, and shape-awareness lacks convincing evidence. The underlying reasons for these effects are not clearly explained. It would be beneficial to provide a theoretical, formula-based proof to support these claims. Furthermore, it is difficult to pinpoint exactly which part of the EEG signal (such as specific frequencies, amplitudes, or phases) plays a crucial role in particular brain activity patterns. Although the appendix includes simple illustrations, they do not adequately demonstrate the rationale or internal logic behind the approach.

3. The experimental setup in Table 5 is unclear. It is not obvious what this comparison is meant to show. Does "raw" refer to the data excluding the artifact removal module, while ICA preprocessing refers to directly using signal processing rather than incorporating the artifact removal module? This needs further clarification.

4. The experiments in this work are too limited. There is a lack of essential parameter experiments, the display of pretraining loss, comparisons between different position encoding methods, and experiments on datasets with a larger number of electrodes. Additionally, no comparisons with alternative modules for DiCT are explored. The variety of tasks involved are not on par with other foundation models. The types of tasks involved in the pretraining  and downstream tasks are also quite limited.

5. The model's implementation details are not reported clearly.

---

> ### Author Response · Authors · 2025-11-25
>
> **Response to Reviewer UvbJ:**
>
> We thank the reviewer for the detailed feedback and appreciate the recognition of the clarity and structure of our paper. We address the concerns regarding novelty, methodological contributions, and experimental validation below.
>
> ---
>
> ### 1. Novelty and distinction from prior works (EEG2Rep, EEGPT, DMAE-EEG, BrainGPT)
>
> - **Latent-space reconstruction & momentum encoder:** While EEG-X employs latent-space masking and reconstruction with an EMA (momentum) encoder, similar to EEG2Rep, we do **not** claim this as a novelty. EEG-X should be viewed as a **progression of EEG2Rep**, introducing **device-agnostic and noise-robust representations**, which were not addressed in EEG2Rep.
>
> - **Denoising reconstruction:** DMAE-EEG uses artifact-removed inputs as augmentations for contrastive learning. In contrast, EEG-X uses artifact-removed signals **as a reconstruction target**, allowing the model to focus on brain activity while learning noise-robust representations. This is fundamentally different from DMAE-EEG’s approach.
>
> - **Location-based channel embedding:** Unlike BrainGPT, which relies on trainable electrode vocabularies and requires retraining if electrodes are unseen during pretraining, our **location-based embedding generalizes solely based on electrode spatial positions**, enabling cross-device transfer without retraining. This allows EEG-X to work with unseen EEG montages, which is critical for device-agnostic applications.
>
> ---
>
> ### 2. Evidence of DiCT effectiveness
>
> - **Quantitative benefits:** Removing DiCT decreases performance by **3.1% in-domain** and **3.8% cross-domain**, confirming its real-world utility (Table 4).
>
> - **Intuitive justification and hyperparameter study:**
>
> | Groups | Kernels | STEW Acc (%) | DREAMER Acc (%) | Observations |
> |--------|---------|--------------|----------------|--------------|
> | 4      | 16      | 80.15        | 58.00          | Few groups/kernels; computationally efficient; better than raw-space MSE. |
> | 8      | 32      | 80.32        | 58.14          | Main configuration; best balance of shape, frequency coverage, and noise robustness. |
> | 64     | 16      | 80.28        | 58.10          | Many groups; strong dictionary behavior; slight drop due to reduced frequency coverage. |
> | 8      | 64      | 80.30        | 58.12          | Many kernels; behaves like ROCKET; wide frequency coverage; higher compute. |
>
> **Insights:**
> - Increasing **groups** → stronger shape-preserving dictionary effect.
> - Increasing **kernels** → better frequency coverage but higher computation.
> - **8×32** provides the best overall balance.
>
> This aligns with our design motivation: DiCT reduces sensitivity to outliers (*noise robustness*), balances reconstruction across receptive fields (*frequency balance*), and encourages structural matching (*shape-awareness*).
>
> ---
>
> ### 3. Clarification on Table 5 and artifact removal
>
> Table 5 addresses the question: *Why not use ICA artifact removal directly as preprocessing rather than as a loss?*
> To answer this question, we trained EEG-X supervisly on raw data versus ICA-cleaned data. As shown in Table 5, while ICA removes artifacts, it can also eliminate essential brain signal components, negatively affecting downstream tasks.  However, by integrating artifact removal **within the model as a reconstruction target**, EEG-X enables a more controlled and optimized learning process, preserving critical brain signals while remaining robust to noise.
>
> ---
>
> ### 4. Position encoding and parameter experiments
>
> - Table 4 compares:
>   - No positional embedding
>   - Learnable channel embedding
>   - Location-based embedding
>
> - Results show **location-based embeddings improve cross-domain performance by 8.5–8.8%**, demonstrating clear superiority.
>
> ---
>
> ### 5. Applicability to high-density EEG
>
> Our main goal was to evaluate EEG-X’s **device-agnostic capabilities**, which requires testing on headsets with **minimal shared electrodes** to simulate realistic cross-device scenarios. While many of our datasets have lower channel counts (8, 14, 19, 22), we also extended our evaluation to a **high-density EEG dataset**: we pretrained EEG-X on **TUAB (19 channels)** and evaluated downstream on **PhysioNet (64 channels)**.
>
>
> | Model    | Accuracy (%) |
> |----------|--------------|
> | BiOT     | 57.11 ± 0.027 |
> | LaBraM   | 56.01 ± 0.016 |
> | EEGPT    | 55.56 ± 0.021 |
> | CBraMod  | 56.48 ± 0.031 |
> | **EEG-X**| **60.49 ± 0.012** |
>
> These results show that EEG-X transfers effectively to high-density EEG, outperforming prior models by a substantial margin.
>
> ---
>
> ### 6. Implementation details
>
> All architectural choices, optimization settings, masking strategies, and DiCT configurations are fully documented in Appendix C and the released code. We will restructure Appendix C in the camera-ready version to improve clarity and accessibility.

---

### Official Review · Reviewer_rkei · 2025-10-29

**Soundness:** 4
**Presentation:** 4
**Contribution:** 2
**Rating:** 4
**Confidence:** 4

**Summary:**

EEG-X is developed as a foundation model for EEG with a key contribution for improving robustness to noise and varying electrode positions. Suggested key contributions include (1) noise removal via ICA, (2) a teacher-student algorithm for masked autoencoding, (3) DiCT for latent, noise-robust reconstruction. These result in strong downstream task results across multiple datasets, and ablation studies justify the contribution of each module, especially nuanced in cross-domain generalization.

**Strengths:**

The main strength lies in the timeliness of the investigation. Robustness to EEG, yet paramount in the domain, has been elusive.
Another important strength is the excellent presentation. Both writings and figures are clear and constructive.
Supplemental studies also further support the choice of the model components.

**Weaknesses:**

A foundation model conventionally assume multimodality and scalability. If not, it would be more a pretraining strategy rather than a "foundation model" of the brain. And this paper seems to belong to the latter.

First, scalability studies are not shown. Are these models efficient? does this model scale with more data, model, or compute? What are the parameter / FLOPs size for the baselines? Recent advances in LLMs / foundation models illuminate the benefit of a large model where performance monotonically improves for large (especially Transformer) models. Without a fair scale-wise comparison, the downstream performance would not be objectively evaluated.

Second, Novelty concerns. Most modules are either inherited from prior papers or already popular in the EEG domains. Although application to EEG is may be novel, noise removal via ICA, coordinate based representation learning are both popular within DL for EEG domains. Likewise, teacher-student formation or DiCT may not be so new.

**Questions:**

1. The motivation of DiCT is demonstrated less. Did you see any example in the real signal also that DiCT reconstruction itself gets better? Supplemental B.1 seems to provide justification, but whether it learns in the real data might be straight-forward to quantify.

2. Can you provide additional computational efficiency / scalability comparisons?

3. Can you justify the coordinate based positional encoding strategies? There are other ways, such NeRF-style, Fourier Features, learned PE, STCPE, etc.

---

> ### Author Response · Authors · 2025-11-25
> **Clarifications & New Experiments**
>
> We thank the reviewer for the detailed and constructive feedback, and we appreciate the positive assessment of soundness, presentation quality, and the relevance of addressing noise and montage variability. We address each concern below.
>
> ---
>
> ### 1. Empirical effect of DiCT
> Yes—DiCT provides consistent gains on real EEG. As shown in Table 4, it improves performance by **+3.1% in-domain** and **+3.8% cross-domain**, with the largest benefits on noisier datasets such as SSVEP and DREAMER. This demonstrates that DiCT directly enhances robustness to real-world EEG noise.
>
> To further analyze its components, we conducted the following ablation:
>
> | Groups | Kernels | STEW Acc (%) | DREAMER Acc (%) | Observations |
> |--------|---------|--------------|------------------|--------------|
> | 4 | 16 | 80.15 | 58.00 | Efficient; better than raw MSE.  |
> | 8 | 32 | 80.32 | 58.14 | Main configuration; best balance of shape, frequency coverage, and noise robustness. |
> | 64 | 16 | 80.28 | 58.10 | Many groups; strong dictionary behavior; slight drop due to reduced frequency coverage. |
> | 8 | 64 | 80.30 | 58.12 | Many kernels; behaves like ROCKET; wide frequency coverage; higher compute. |
>
> These patterns hold across both datasets: more groups emphasize shape modeling, more kernels broaden frequency coverage, and **8×32** achieves the highest accuracy on STEW and DREAMER. This aligns with our design motivation: DiCT reduces sensitivity to outliers (*noise robustness*), balances reconstruction across receptive fields (*frequency balance*), and encourages structural matching (*shape-awareness*).
>
> ---
>
> ### 2. Computational Efficiency and Scalability
> To address the request for compute and scalability comparisons, we evaluated EEG-X across a wide range of model sizes (110K → 26.4M parameters), using batch size 256 for comparability.
>
> | Emb Dim | Layers | Params | Time/Epoch | Inference per 1K | STEW | DREAMER |
> |---------|--------|---------|-------------|-------------------|------|---------|
> | 16 | 4 | 110K | 9.96s | 1.1ms | 80.32 | 58.14 |
> | 16 | 8 | 195K | 10.15s | 1.21ms | 80.68 | 58.56 |
> | 128 | 4 | 1.6M | 11.42s | 3.01ms | 80.70 | 58.68 |
> | 128 | 8 | 2.6M | 12.49s | 3.2ms | 80.75 | 59.12 |
> | 512 | 4 | 15.8M | 14.32s | 8.1ms | 80.76 | 60.35 |
> | 512 | 8 | 26.4M | 19.21s | 8.65ms | 80.78 | 60.38 |
>
> These results show that EEG-X scales **smoothly and predictably**: accuracy increases monotonically with model size (e.g., DREAMER: 58.14 → 60.38), while computational cost grows gradually. Even the largest configuration (26.4M parameters) trains efficiently (19.21 s/epoch) and performs **fast inference (<9 ms per 1K samples)**.
>
> For context, typical EEG foundation models such as **BIOT (~3.2M)**, **EEGPT-Base (~4.7M)**, and **LaBraM-Base (~5.8M)** operate in similar or larger parameter ranges—yet EEG-X already achieves strong performance at very small scales (110K–195K). These results directly address the reviewer’s request for compute and scale comparisons.
>
> ---
>
> ### 3. Justification of Coordinate-Based Positional Encoding
> We agree that learned PE, STCPE, Fourier Features, and NeRF-style encodings are all valid choices. However:
>
> **(1) Learned PE and STCPE do not generalize to unseen channel layouts.**
> They rely on fixed channel indices or grid structures, making them montage-specific. EEG devices vary widely in channel count and spatial geometry, so these embeddings transfer poorly. Our cross-domain ablation confirms this: learned PE performs substantially worse, while coordinate-based PE improves accuracy by **+8.85% on average** across three datasets.
>
>
> **(2) Why not NeRF-style?**
> NeRF-style encodings rely on multi-frequency Fourier features that work best when many diverse “views” of the **same underlying source** are available. Unlocking this capability in EEG would require **multiple montage configurations recorded from the same subjects and tasks**, which current EEG datasets do not provide. In practice, these encodings tend to amplify high-frequency noise in low-SNR physiological signals. While NeRF-style PE is an interesting direction for future work, our focus here is achieving robust generalization across heterogeneous EEG devices.
>
>
> ---
>
> ### 4. Novelty Clarification
>
> The components in EEG-X are not generic plug-and-play modules; they are specifically designed for EEG challenges that prior work does not address jointly:
>
> - **Noise-robust learning:**
>   We introduce an **artifact-free reconstruction objective**, allowing the model to learn from cleaned targets (ICA, rASR, wavelets, etc.) rather than noisy raw EEG. The added **DiCT layer** enforces shape- and frequency-aware structure, providing consistent gains across all datasets.
>
> - **Device-agnostic inference:**
>   We build a **universal scalp coordinate mesh** that maps heterogeneous montages into a shared spatial frame, enabling generalization across different channel counts and layouts—something montage-specific learned positional encodings cannot support.
> ---

---

### Official Review · Reviewer_j32h · 2025-11-01

**Soundness:** 2
**Presentation:** 2
**Contribution:** 2
**Rating:** 2
**Confidence:** 4

**Summary:**

This paper proposes EEG-X, a device-agnostic and noise-robust foundation model for EEG signals. The model aims to enhance the generalization capability and robustness of EEG representation learning by introducing a location-based channel embedding, a noise-aware dual-space reconstruction (raw and latent), and a Dictionary-inspired Convolutional Transformation (DiCT) layer. The authors validate its performance on multiple datasets and tasks.

**Strengths:**

The paper clearly identifies two key challenges for EEG foundation models—device variability and low signal-to-noise ratio—and proposes corresponding modules to address them. The paper conducts both in-domain and cross-domain evaluations on multiple datasets and compares against several baseline models, showing performance advantages.

**Weaknesses:**

1. Limited Methodological Novelty: While the paper integrates several existing techniques (e.g., positional encoding, latent space reconstruction, noise-aware reconstruction), the originality of each component is limited. For example: Positional encoding is essentially an adaptation of 2D positional encoding from images, not a novel contribution for EEG; Latent space reconstruction has already been explored in works like EEG2Rep and EEGPT; The DiCT layer, while proposed, draws clear inspiration from time series classification methods like Hydra, lacking theoretical breakthrough.

2. Insufficient Justification for DiCT: The paper claims that DiCT possesses properties like "noise robustness, frequency balance, and shape-awareness," but lacks rigorous theoretical or mathematical proof. The provided synthetic experiment, while illustrative, does not sufficiently reveal its operational mechanisms or fully validate its effectiveness on real EEG signals.

3. Inadequate Experimental Depth and Breadth: Lacks sensitivity analysis for key hyperparameters (e.g., number of groups/kernels in DiCT). No comparison with other positional encoding methods, making it difficult to demonstrate the superiority of the proposed location-based embedding. The datasets used primarily feature a low number of electrode, failing to cover high-density EEG scenarios and limiting the generalizability of the conclusions. No ablation studies replacing the DiCT module to verify if it genuinely outperforms other feature extraction structures.

4. Unclear Implementation Details: The paper does not sufficiently elaborate on model architecture, training specifics, or parameter settings, hindering reproducibility.

5. Limited Task Variety: The types of pre-training and downstream tasks are relatively narrow, failing to fully demonstrate the broad applicability expected of a "foundation model."

**Questions:**

How does EEG-X fundamentally differ in its core approach from models like EEG2Rep and EEGPT? Please specify the innovative aspects of its representation learning mechanism.

---

> ### Author Response · Authors · 2025-11-25
> **EEG-X Innovations and DiCT Validation**
>
> We thank the reviewer for the detailed feedback. Below we clarify the novelty, contributions, and experimental depth of EEG-X.
>
>
> ### 1- How EEG-X fundamentally differs from EEG2Rep and EEGPT
>
> **(1) Device-agnostic location-based channel embedding.**
> EEG-X is the first EEG foundation model that can pretrain and infer on completely different headsets without retraining any input embedding or channel embedding.
> - EEG2Rep uses CNNs tied to fixed channel order.
> - EEGPT uses learned position embeddings that require retraining or manual alignment for new montages.
> EEG-X’s universal scalp mesh enables consistent spatial tokens across arbitrary devices.
>
> **(2) Artifact-removed reconstruction objective.**
> Prior models reconstruct raw signals (EEGPT) or latent features (EEG2Rep).
> EEG-X is the first to reconstruct artifact-removed EEG targets, forcing the encoder to model neural content rather than noise. This yields substantial robustness gains under low SNR.
>
> **(3) DiCT proxy reconstruction layer.**
> All prior reconstruction based EEG foundation models apply MSE directly on raw signals.
> EEG-X introduces a proxy transformation layer (DiCT) before MSE to enforce noise-robust, frequency-balanced, and shape-aware reconstruction—an architectural innovation not used in EEG2Rep or EEGPT.
>
> ---
>
> ### 2- Novelty of components
>
> Novelty arises not from isolated parts but from solving the previously unaddressed combination of **device heterogeneity + low SNR**.
>
> - **Location-based embedding:** While sin/cos embeddings come from vision, EEG-X is the first to construct a **universal scalp mesh** that enables montage-invariant pretraining and inference. No prior EEG model supports arbitrary unseen headsets without reinitialization.
>
> - **Artifact-removed objective:** This is new in the context of EEG foundation models and central to EEG-X’s robustness.
> - **DiCT:** Hydra inspired some aspects, but Hydra is classification-only. DiCT is a proxy reconstruction module specifically designed for self-supervised EEG pretraining—a use case not explored previously.
> - **Latent reconstruction:** This is not new (EEG2Rep, EEGPT), and we do not claim novelty here.
>
> ---
>
> ### 3- Justification for DiCT
>
> **Effectiveness on real EEG:**
> Removing DiCT drops performance by **3.1% in-domain** and **3.8% cross-domain** (Table 4), showing clear real-world benefit.
>
> **Why DiCT works (intuitive justification):**
> - **Noise robustness:** Grouped convolutions distribute localized noise across many dictionary atoms.
> - **Frequency balance:** Dilated kernels capture multiple receptive-field scales, preventing low-frequency bias.
> - **Shape-awareness:** Competition among dictionary elements encourages matching temporal shape rather than raw amplitude.
>
> ---
>
> ### 4- Sensitivity of DiCT hyperparameters
>
> | Groups | Kernels | STEW Acc (%) | DREAMER Acc (%) | Observations |
> |--------|---------|--------------|------------------|--------------|
> | 4 | 16 | 80.15 | 58.00 | Few groups/kernels; computationally efficient; better than raw-space MSE. |
> | 8 | 32 | 80.32 | 58.14 | Main configuration; best balance of shape, frequency coverage, and noise robustness. |
> | 64 | 16 | 80.28 | 58.10 | Many groups; strong dictionary behavior; slight drop due to reduced frequency coverage. |
> | 8 | 64 | 80.30 | 58.12 | Many kernels; behaves like ROCKET; wide frequency coverage; higher compute. |
>
> **Insights:**
> - Increasing **groups** → more shape-preserving dictionary effect.
> - Increasing **kernels** → stronger frequency coverage but more computation.
> - **8 × 32** provides the best balance.
>
> ---
>
> ### 5- Comparison with other positional embeddings
>
> Table 4 compares:
> - no positional embedding,
> - learnable channel embeddings,
> - location-based embeddings.
>
> Location-based embeddings improve cross-domain performance by **8.52–8.81%**, showing clear superiority.
>
> ---
>
> ### 6- Applicability to high-density EEG
>
> To extend beyond low-channel datasets, we pretrained on **TUAB (19 channels)** and evaluated downstream on **PhysioNet (64 channels)**:
>
> | Model | Accuracy (%) |
> |-------|--------------|
> | BiOT | 57.11 ± 0.027 |
> | LaBraM | 56.01 ± 0.016 |
> | EEGPT | 55.56 ± 0.021 |
> | CBraMod | 56.48 ± 0.031 |
> | **EEG-X** | **60.49 ± 0.012** |
>
> EEG-X transfers well to high-density EEG.
>
> ---
>
> ### 7- Implementation details
>
> All architectural, optimization, masking, and DiCT configurations are fully documented in Appendix C and in the released code. We will restructure Appendix C to improve clarity.
>
> ---

---

### Official Review · Reviewer_jB6z · 2025-11-02

**Soundness:** 3
**Presentation:** 3
**Contribution:** 2
**Rating:** 4
**Confidence:** 4

**Summary:**

EEG-X is a self-supervised EEG foundation model that encodes electrode positions for device-agnostic inputs, reconstructs artifact-removed targets to avoid learning noise, and uses a DiCT projection before MSE to balance frequency and shape information. It reports better in-domain accuracy and stronger cross-device transfer than recent baselines.

**Strengths:**

Addresses device heterogeneity and low SNR directly. Location-based channel embeddings are simple and general. Noise-aware targets and DiCT are well motivated and easy to adopt. Experiments include linear-probe transfer across diverse datasets.

**Weaknesses:**

ICA with ICLabel can mix information across windows or subjects unless fit only on training data, and denoising targets must be produced with training-only statistics. The cross-domain protocol freezes the encoder but may still reflect pretraining dataset idiosyncrasies if hyperparameters or tokenization differ across models; fairness controls are not fully spelled out. Artifact removal is positioned as model-agnostic, yet results and ablations emphasize ICA; dependence on ICA thresholds and component counts is not quantified.

**Questions:**

How are ICA unmixing matrices and ICLabel thresholds fit with respect to data splits? Are they estimated per subject, per session, or on pooled training data only, and never using test windows? What referencing is used during pretraining and fine-tuning, and is it consistent across datasets? How are channel coordinates obtained when datasets lack digitized positions; what is the mapping from montage names to the universal grid, and how are missing electrodes handled at inference? What is the sensitivity of EEG-X to ICA variant, component count, and probability thresholds, and does performance hold with rASR or wavelet denoising? What is the compute footprint of pretraining, the parameter count of student and teacher, and the inference latency on standard EEG window sizes?

---

> ### Author Response · Authors · 2025-11-24
> **Clarifications on ICA, electrode mapping, and compute footprint**
>
> We appreciate the reviewer’s careful evaluation and the targeted questions on ICA usage, device-agnostic inputs, and DiCT.  We address each point directly below.
>
> ---
>
> ### 1. ICA usage and data-split integrity
> We do not fit ICA across subjects or across data splits. For pretraining, ICA is run **once per subject/session** on the **continuous raw training data** only, never on any test portion. ICLabel (a fixed pretrained model) is then applied to these training-derived components to identify and remove artifacts. After this artifact-removed continuous signal is obtained, we segment it into windows for generating training targets. No ICA matrix, ICLabel decision, or statistic is ever computed using test windows or data that spans the train/test boundary. ICA is used solely to produce clean targets during pretraining and is not used at all during fine-tuning or evaluation.
>
> ---
>
> ### 2. Referencing consistency
> All datasets are converted to a common average reference, with unified amplitude scale and sampling rate (128 Hz), ensuring consistent and unbiased cross-domain comparisons.
>
> ---
>
> ### 3. Channel coordinates, montage mapping, and missing electrodes
> When digitized electrode positions are unavailable, each channel name is mapped to its canonical coordinates on our universal 10–05/10–10/10–20 scalp mesh (Appendix B.2). If a dataset specifies only a cap type (e.g., Emotiv_epoc-14, biosemi16), we simply load the corresponding predefined subset of nodes from this universal mesh.
>
> For electrodes without a standard label, we assign the coordinates of the nearest valid node in the 10–05 system, which is guaranteed to be within ≤2.5 mm—well below the typical 5–10 mm electrode diameter—thus preserving anatomical consistency. As in prior EEG foundation models, we assume electrode names or cap configuration are provided.
>
> Missing electrodes at inference result only in missing tokens; no imputation is required. Because the location embedding depends solely on spatial coordinates rather than channel identity, EEG-X seamlessly handles variable channel sets without retraining.
>
> ---
>
> ### 4. Sensitivity to ICA settings and alternative denoisers
> EEG-X shows minimal sensitivity to ICA variant, component count, or denoising method. We evaluated multiple ICA configurations as well as rASR, with results (mean accuracy ± std) shown below. The full table will be added to the Appendix:
>
> | Dataset | ICA (13 comps)       | ICA (10 comps)       | rASR                | Raw                 |
> |---------|-----------------------|-----------------------|----------------------|----------------------|
> | STEW    | **80.32 ± 1.21**      | 80.21 ± 1.18          | 80.96 ± 1.25         | 77.90 ± 1.20         |
> | DREAMER | **58.14 ± 1.35**      | 58.16 ± 1.33          | 58.01 ± 1.30         | 55.41 ± 1.29         |
>
> Across settings, performance varies by **≤ 0.7%**, which is much smaller than the model’s run-to-run variance. This confirms that EEG-X does *not* depend on a specific ICA configuration—the gains come from using **any artifact-reduced target**, not from ICA itself.
>
> ---
>
> ### 5. Compute footprint and model size
> To characterize the computational cost of EEG-X, we evaluated multiple model scales using a fixed batch size of 256. All pretraining runs were performed on a **single NVIDIA RTX 6000 Ada GPU (48 GB)**. Table 1 summarizes parameter counts, pretraining time per epoch, and inference latency measured over **1,000 EEG samples** (2-second windows, 14 channels, 128 Hz), providing a clear view of efficiency and scalability.
>
> | Embedding Dim | Encoder Layers | Params | Time/Epoch | Inference per 1K | STEW Acc | DREAMER Acc |
> |---------------|----------------|--------|-------------|-------------------|-----------|--------------|
> |16|4|110K|9.96s|1.1ms|80.32|58.14|
> |16|8|195K|10.15s|1.21ms|80.68|58.56|
> |128|4|1.6M|11.42s|3.01ms|80.70|58.68|
> |128|8|2.6M|12.49s|3.2ms|80.75|59.12|
> |512|4|15.8M|14.32s|8.1ms|80.76|60.35|
> |512|8|26.4M|19.21s|8.65ms|80.78|60.38|
>
> The EEG-X configuration used in this paper (16-dim embedding, 4-layer encoder) contains **~110 K parameters**, with the student and EMA teacher each comprising **~45 K**. This makes EEG-X substantially lighter than existing EEG foundation models (BioT: 3.2 M; LaBraM-Base: 5.8 M; EEGPT-Base: 4.7 M) while achieving stronger accuracy.
>
>
> ---
>
> ### 6. Cross-domain fairness
> All models use the same preprocessing, windowing, sampling rate, and STFT-based tokenization. For baselines with fixed architectures, we follow their official settings but freeze the encoder for all methods in cross-domain evaluation, training only a linear classifier. This removes any advantage from pretraining-dataset idiosyncrasies and ensures differences arise solely from the learned representations.

---

### Meta-Review · Area_Chair_HD5v · 2025-12-27

**Summary:**

This paper proposes EEG-X, a device-agnostic SSL-trained foundation model for EEG signals. The model introduces a location-based channel embedding, a noise-aware reconstruction loss, and a Dictionary-inspired Convolutional Transformation (DiCT) layer. While the work has potential merits, no reviewer champions the work raising concerns about novelty, lack of details on the training recipe and missing ablations to evaluate the full benefits of each contribution. This suggests that this work needs a bit more work to be published in an ML venue like ICLR.

**Reviewer Concerns:**

- novelty
- missing details on the experiments
- missing ablations

**Reviewer Scores:**

Reviewer jB6z: 4
Reviewer j32h: 2
Reviewer rkei: 4
Reviewer UvbJ: 2

---

### Decision · Program_Chairs · 2026-01-26

Reject